# Rigid Motion Compensated Compressed Sensing MRI with Untrained Neural Networks

## Abstract

Deep neural networks trained end-to-end for accelerated magnetic resonance imaging give excellent performance. Typically, these networks are trained and evaluated under a setup where the object to be imaged is static. However, in practice, patients often move during data acquisition which leads to motion artifacts in the reconstructed images. In this work, we first demonstrate that in the presence of motion, significantly larger training sets are required for good performance when training state-of-the-art neural networks to reconstruct an image for accelerated MRI. Second, we demonstrate that as an alternative, one can resort to utilizing untrained neural networks for this task. We propose a modified untrained network which does not rely on any training set and performs single-instance rigid motion-compensated compressed sensing MRI. Our approach outperforms untrained and trained optimization-based baselines such as $\ell_1$-norm minimization and score-based generative models.

## 1 Introduction

Deep learning methods give state-of-the-art performance for many image restoration applications (Dong et al., 2014; Jin et al., 2017; Zhang et al., 2017; Sriram et al., 2020; Rivenson et al., 2018; Jalal et al., 2021; Zhang et al., 2023), including for accelerated MRI reconstruction where the goal is to reconstruct a high-quality MRI scan from a set of undersampled measurements. Most successful deep learning-based accelerated MRI reconstruction models assume a static imaging setup, meaning that a potential patient movement is not anticipated. Consequently, in case the patient moves during data acquisition, motion artifacts arise and the image quality significantly degrades.

One possible approach to deal with motion artifacts is to simply train a network to reconstruct motion-corrupted data. In this work, we first investigate this avenue, and find that motion-compensated accelerated MRI reconstruction is very costly in terms of the amount of data required for training. Thus, switching the task from artifact-free to motion-compensated accelerated MRI reconstruction brings a significant burden in terms of the amount of data to be collected to train state-of-the-art MRI models.

Subsequently, we propose to resort to untrained neural networks as an alternative. These models operate in a single-instance reconstruction mode and do not require a large training set. We propose an untrained network based on the ConvDecoder (Zalbagi Darestani & Heckel, 2021), an untrained network tailored to MRI reconstruction. We specifically modify ConvDecoder's loss function to handle motion correction in addition to compressed sensing.

To summarize, here are our contributions:

- We demonstrate that state-of-the-art MRI reconstruction models require significantly more data than the currently available large training sets in order to solve motion correction and compressed sensing MRI at the same time.
- We propose an untrained network-based approach to perform motion-compensated accelerated MRI reconstruction.
- We evaluate our approach for 2D and achieve competitive performance against other baselines such as sparsity-based and score-based models. Furthermore, proof of principle is also demonstrated for 3D MRI data.

## 1.1 PRIOR WORK

Over the past few years, several works have tackled the problem of motion artifact correction in MRI using prospectively or retrospectively deep learning approaches. In general, one may categorize those works as follows:

**Model-based:** These methods typically solve an optimization problem for each input sample by incorporating knowledge of the physical measurement model (i.e., the forward operator $\mathbf{A}$). In order to perform motion correction, optimization is often done with respect to two sets of variables, one parameterizing the image and one for the motion parameters. After convergence, the outputs are estimates of the ground-truth image and motion parameters. Sparsity-based methods fall under this category (Reyes et al., 2007; Yang et al., 2013; Mayer et al., 2022).

**Data-driven:** Several end-to-end deep learning-based models have made efforts to solve the motion correction problem by training a neural network to learn a mapping from the motion-corrupted image domain to the artifact-free image domain (Pawar et al., 2018; Al-Masni et al., 2022). These models typically ignore the forward model and tackle the problem in a data-driven manner. A major limitation of data-driven approaches is that reconstructed images tend to be blurry (this is an observation we made for U-Net (Ronneberger et al., 2015) and E2E-VarNet (Sriram et al., 2020) but is also seen in several other works (Pawar et al., 2018; Armanious et al., 2020)).

**Data-driven and model-based:** These methods tend to combine deep learning with model-based optimization in order to correct motion artifacts. For example, Hossbach et al. (2022) trained a neural network to predict motion parameters from the data, and then used those predictions as an initialization for a sparsity-based method to correct motion artifacts. Score-based generative models are also an example of this category. They rely on a pre-trained generator that is used inside an optimization problem at inference. In this manner, they are claimed to be more robust against variable motion patterns Levac et al. (2022). Score-based generative models also outperform traditional generative models for medical imaging (Armanious et al., 2020).

## 2 PROBLEM SETUP: MOTION CORRUPTED COMPRESSED SENSING

Our goal is to reconstruct an image $\mathbf{x}^* \in \mathbb{C}^N$ from undersampled measurements $\mathbf{y} = \mathbf{MFTx}^* + \mathbf{z} \in \mathbb{C}^M$, where the number of measurements, $M$, is typically lower than the dimension of the image, $N$, and $\mathbf{z}$ is measurement noise. In the forward map, $\mathbf{M}$ is the known undersampling mask, $\mathbf{F}$ is the Fourier transform, and $\mathbf{T}$ denotes the unknown rigid motion transform discussed in more detail below. The measurement $\mathbf{y}$ is usually called the $k$-space in the context of MRI.

In practice, multiple receiver coils are used for signal reception, so there are $n_c$ coils each capturing a $k$-space measurement with an at least a slightly different spatial sensitivity profile. Thus, there are $n_c$ many $k$-spaces obtained as

$$\mathbf{y}_i = \mathbf{MFTS}_i\mathbf{x}^* + \mathbf{z}_i \in \mathbb{C}^M, \quad i = 1, \ldots, n_c.$$

Here, $n_c$ denotes the number of receiver coils, $\mathbf{S}_i$ is the complex-valued spatially-varying coil-dependent sensitivity map of the $i$-th coil, that is applied through element-wise multiplication to the image $\mathbf{x}^*$, and $\mathbf{z}_i$ is measurement noise.

### 2.1 MOTION ARTIFACT SYNTHESIS

We now specify the assumptions we make on the unknown motion transform $\mathbf{T}$. Assuming a model for the motion transform is important for our study, since patient movements are naturally unknown, and thus one needs to make certain assumptions about these motion patterns in practice.

There are in general two types of motion occurring during an MRI scan: rigid motion and nonrigid motion. Rigid motion results in linear transformations in the image and is typically caused by translations or rotations in 3D (e.g., head movements). Nonrigid motion results in anatomical deformations in the scanned image and is typically caused by non-shape-preserving object transformations (e.g., respiratory motion).

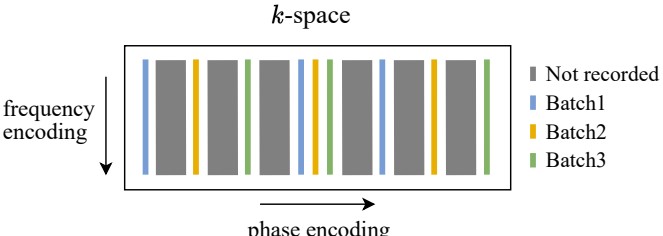

Figure 1: **An example of interleaved trajectory with equispaced undersampling.** In this example, there are 3 repetition times (TRs) corresponding to 3 batches with 3 acquired lines per batch. This means that for instance $k$-space lines corresponding to the 3 blue lines in the trajectory are recorded during the first repetition time.

In this work, we primarily consider rigid motion caused by 2D translations. However, to demonstrate that our approach is easily applicable to more complicated motion models (i.e., also including rotations), we provide experimental results for 3D motion as well.

For 2D motion synthesis, we consider an interleaved trajectory with a 1D equispaced undersampling pattern (with a fully-sampled center region), see Figure 1 for an example. We synthesize translation artifacts by a simple linear phase shift in the $k$-space. Specifically, the $k$-space pixel value at coordinates $(x, y)$ is transformed as follows under $(t_x, t_y)$ translations along the x and y axes:

$$\tilde{\mathbf{k}}_{xy} = \mathbf{k}_{xy} * e^{2\pi j(t_x x + t_y y)}.$$

Note that all $k$-space lines acquired during a given repetition time (TR) are, in a first approximation, assumed to be acquired instantaneously, and thus these lines are affected by the same transformation. Therefore, t number of x- and y-axis translation coefficients form the motion transform (t is the number of TRs). From this point onward, we denote a motion transform as $\mathbf{T}_\phi$ where $\phi \in \mathbb{R}^{2t}$ contains all translation parameters. For experiments with 3D data, $\phi \in \mathbb{R}^{6t}$ models 6 degrees of freedom which are $(t_x, t_y, t_z)$ translations and $(\alpha, \beta, \gamma)$ rotations.

## 3 END-TO-END NETWORKS ARE COSTLY FOR MOTION-COMPENSATED COMPRESSED SENSING MRI

Neural networks trained end-to-end give state-of-the-art accuracy for accelerated MRI reconstruction for a static setup, i.e., for a setup where the patient does not move. Thus, a natural starting point to develop a neural network for motion-compensated accelerated MRI is to train a neural network end-to-end for reconstruction from motion-corrupted data. In this section, we demonstrate that training a neural network end-to-end for motion-compensation is very expensive in the number of training examples required.

We consider the popular class of unrolled networks, the best-performing networks for accelerated MRI reconstruction (Sriram et al., 2020; Fabian & Soltanolkotabi, 2022). The idea behind these models is to unroll an optimization problem and learn several iterates of it in an end-to-end manner. Here, we study the end-to-end variational network architecture (Sriram et al., 2020) (E2E-VarNet). For motion-corrupted accelerated MRI reconstruction, we modify each cascade of the E2E-VarNet's from

$$\mathbf{k}^{i+1} = \mathbf{k}^i - \eta(\mathbf{M}\mathbf{k}^i - \mathbf{y}) + G(\mathbf{k}^i)$$

to

$$\mathbf{k}^{i+1} = \mathbf{k}^i - \eta(\mathbf{M}\mathbf{T}_\phi \mathbf{k}^i - \mathbf{y}) + G(\mathbf{k}^i), \tag{1}$$

in order to accound for the change in forward map. Note that only the data consistency block is modified by incorporating the motion transform $\mathbf{T}_\phi$. Here, $G : \mathbb{R}^n \to \mathbb{R}^n$ is a trainable neural network (i.e., the learned regularizer) which performs refinement by mapping the current estimate of the $k$-space to a refined $k$-space estimate for the next step. In this setup, the parameters of network $G$ and the parameters of a network that learns motion parameters $\phi$ are trained.

To evaluate the potential performance of this modified E2E-VarNet, we conduct the following experiment. We assume that motion parameters (i.e., $\phi^*$) are perfectly known during training and inference. This is an idealized situation since in practice the motion parameters are unknown and have to be estimated. However, studying this idealized situation clarifies whether this natural extension of a state-of-the-art approach is capable of accurate image recovery for joint motion correction and compressed sensing.

**Experiments.** We use the 2D-recorded multi-coil brain T2 portion of the fastMRI dataset (Zbontar et al., 2020). We created a validation/test split of 160/300 slices. For the training dataset, depending on the setup, we use a total of 850/3400/7587/21296/63888 training samples.

To vary the training set size, we compare two cases: one where we add additional slices from the fastMRI dataset, and one where we keep the number of slices fixed but augment the dataset with more motion patterns. For motion synthesis, we sample x and y translation parameters from a uniform distribution $t_x, t_y \sim \text{Unif}(5, 10)$ according to the model from Section 2.1. Finally for undersampling, we work with a 1D equispaced variable density mask (with 4x acceleration) which is the same for all training and inference samples.

Figure 2 shows the result. Augmenting the training set with more slices (and not with more motion patterns) improves reconstruction accuracy according to a power law. The improvement as a function of training examples does not saturate in the span of the training set sizes that we consider. Contrary, without motion corruption (i.e., the artifact-free regime) we are already in a regime of the power law where only minimal performance improvements occur. The artifact-free power law is consistent with that established for clean (without motion corruption) accelerated MRI reconstruction (Klug & Heckel, 2023). This demonstrates that in order to train a network for motion-corrupted reconstruction, we need a significantly larger dataset size for good performance, even in an ideal setup where we know the motion corruption pattern.

Finally, note that according to Figure 2, a network trained on $\approx$60,000 images achieves 0.92 SSIM for motion-compensated accelerated MRI reconstruction. However, in the artifact-free regime (i.e., when no motion appears during training/inference), the same performance is obtainable by training the same network on only 1000 images. This demonstrates that motion-compensated accelerated MRI reconstruction via E2E-VarNet is much more costly than solving artifact-free accelerated MRI reconstruction.

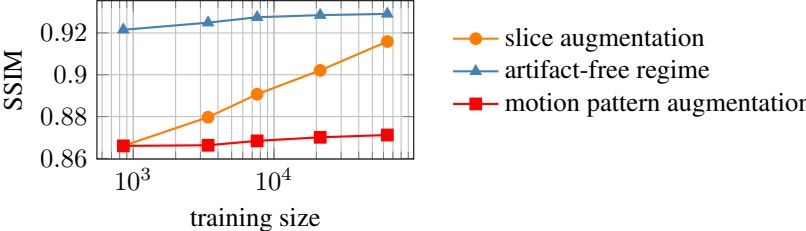

Figure 2: **Test accuracy as a function of training set size.** —●—: Increasing the training set size by adding more slices to the training set. —■—: Increasing the training set size by adding more motion patterns to a fixed set of slices. —▲—: Increasing the number of slices in the artifact-free regime (i.e., reconstruction from clean undersampled data). By comparing the —●— and —▲— curves, the test accuracy scales differently based on the number of training slices which demonstrates the excessive cost of motion-compensated compressed sensing MRI.

With respect to reconstruction quality, Figure 3 shows reconstructions for the experiment above. Note that the reconstruction becomes blurry whenever the input sample is corrupted with motion artifacts and this starts to alleviate with more training examples.

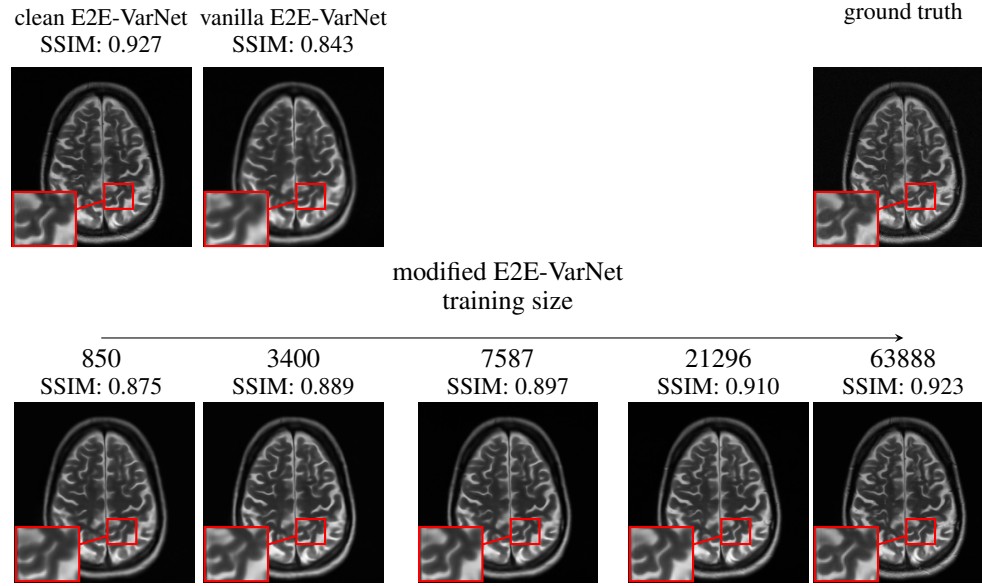

Figure 3: **Quality of modified E2E-VarNet reconstruction from motion-degraded undersampled measurements improves significantly with more training data points. clean E2E-VarNet** is a network that is trained on 850 clean 4x undersampled slices and is applied to a clean test sample (this is the best reconstruction E2E-VarNet can achieve for this test sample). **vanilla E2E-VarNet** is a network that is trained on 850 motion-degraded 4x undersampled slices and is applied to a motion-degraded test sample. **modified E2EVarNet** is a network with a modified DC block for motion correction and is trained on motion-degraded 4x undersampled data, then applied to a motion-degraded test sample. Our modified E2E-VarNet is trained on 850, 3400, 7587, 21296, and 63888 motion-degraded training slices.

# 4 UNTRAINED NETWORKS FOR MOTION-COMPENSATED COMPRESSED SENSING

We propose an approach for motion compensated accelerated MRI based on *untrained neural networks*. Without any training, convolutional neural networks (CNNs) can regularize inverse problems as first demonstrated by (Ulyanov et al., 2018). Untrained network perform well for general compressive sensing tasks (Veen et al., 2018; Heckel & Hand, 2019), and in particular for accelerated MRI reconstruction (Arora et al., 2020; Zalbagi Darestani & Heckel, 2021; Slavkova et al., 2022). Untrained networks outperform traditional untrained methods (such as $\ell_1$-regularized least squares) but perform worse than state-of-the-art MRI reconstruction models such as unrolled neural networks (e.g., the VarNet for static accelerated MRI).

In a nutshell, an untrained network reconstructs an image by fitting a randomly initialized neural network to a measurement. The network is not pretrained on any training data, and the structure of the network alone acts as a prior for the images. Note that for a given task, a few images from the target domain are required only to tune the hyper-parameters of the network.

Although untrained CNNs are successful tools for various image restoration tasks (Ulyanov et al., 2018; Veen et al., 2018; Heckel & Hand, 2019; Jin et al., 2021; Arora et al., 2020; Zalbagi Darestani & Heckel, 2021; Jagatap & Hegde, 2019; Heckel, 2019), they have not yet been explored for image reconstruction from motion-corrupted undersampled data. Here, we propose a variant of the ConvDecoder (Zalbagi Darestani & Heckel, 2021) whose loss function is adjusted to handle motion correction in addition to compressed sensing.

## 4.1 METHOD

Let $G : \mathbb{R}^p \to \mathbb{R}^n$ be a neural network parameterized by $\boldsymbol{\theta} \in \mathbb{R}^p$, specifically we use the convolutional decoder architecture from (Zalbagi Darestani & Heckel, 2021). Given a measurement $\mathbf{y}$ we minimize the loss

$$\mathcal{L}(\boldsymbol{\theta}, \boldsymbol{\phi}) = \|\mathbf{MFT}_{\boldsymbol{\phi}}\mathbf{S}G(\boldsymbol{\theta}) - \mathbf{y}\|_2^2 \tag{2}$$

with gradient descent starting from a random initialization of the network's parameters and zero initialization of the motion parameters. Note that we are optimizing jointly over the networks' parameters, and thus over different images, as well as over the motion parameters, thus over different forward maps.

This optimization yields the estimate $\hat{\boldsymbol{\theta}}$ of the network's parameters, and with this estimate we reconstruct the ground truth image as $\hat{\mathbf{x}} = G(\hat{\boldsymbol{\theta}})$.

The network $G$ we use throughout is based on (Zalbagi Darestani & Heckel, 2021) tuned on 10 randomly-selected samples from the training set of the fastMRI brain dataset (Zbontar et al., 2020). Specifically, the network is a convolutional network with 8 layers and 64 channels per layer. Each convolutional layer comprises upsampling, convolution, ReLU activation, and batch normalization (Ioffe & Szegedy, 2015) blocks. Finally, we use ESPIRiT (Uecker et al., 2014) to estimate coil sensitivity maps $\mathbf{S}$ from the motion-degraded undersampled measurement.

Note that because the sensitivity maps are obtained from the corrupted undersampled data, they are prone to an error caused by patient movements. We therefore assume mild patient movements (which is often the case in practice), and thus the error in the coil sensitivity estimates becomes negligible.

## 4.2 EXPERIMENTS

We evaluate our approach for 2D and 3D motion correction tasks in the following two subsections, respectively.

### 4.2.1 2D MOTION-COMPENSATED COMPRESSED SENSING MRI

Here, we conduct evaluations on 336 middle slices of AXT2 volumes from the validation portion of the fastMRI multicoil brain dataset (Zbontar et al., 2020). Each $k$-space in the dataset we consider has the shape (#coils, 640, 320) with an undersampling ratio of 4; thus 80 out of 320 lines in the $k$-space are recorded. We compare our method with the score-based generative model proposed by (Levac et al., 2022) and $\ell_1$-norm wavelet regularized least-squares.

For motion artifact synthesis, we follow our approach detailed in Section 2.1. Specifically, we first corrupt the $k$-space with motion transform $\mathbf{T}_{\phi^*}$ to obtain a measurement $\mathbf{y}$ of size (#coils, 640, 320), and then undersample the measurement with a factor of 4 using a 1D equispaced variable density mask. Note that three quarters of the 320 vertical lines in $\mathbf{y}$ are now equal to zero due to undersampling.

As for the motion pattern and trajectory of sampling, we consider three settings:

1. 10 TRs and random $x$ and $y$ translations $t_x, t_y \sim \text{Unif}(-2, 2)$ which results in the ground-truth motion parameter $\phi^* \in \mathbb{R}^{10*2}$. This means every 8 lines in the $k$-space are affected by the same motion state.

2. 24 TRs and random $x$ and $y$ translations $t_x, t_y \sim \text{Unif}(-2, 2)$ which results in the ground-truth motion parameter $\phi^* \in \mathbb{R}^{24*2}$.

3. 10 TRs and $x$ and $y$ translations $t_x$ and $t_y$ which results in the ground-truth motion parameter $\phi^* \in \mathbb{R}^{10*2}$. $t_x$ and $t_y$ are generated using sine and cosine functions to create a more realistic motion pattern in the sense that two consecutive motion states are very close to each other.

Table 1 shows the results averaged over 336 slices. The ranking of the methods is Ours > score-based model > $\ell_1$-minimization and this is observed for various types of motion patterns. Figure 4 illustrates reconstruction examples along with motion parameter plots for each method[1]. Looking at those

---

[1]Results of the score-based model are obtained by reproducing the code provided by the authors (Levac et al., 2022).

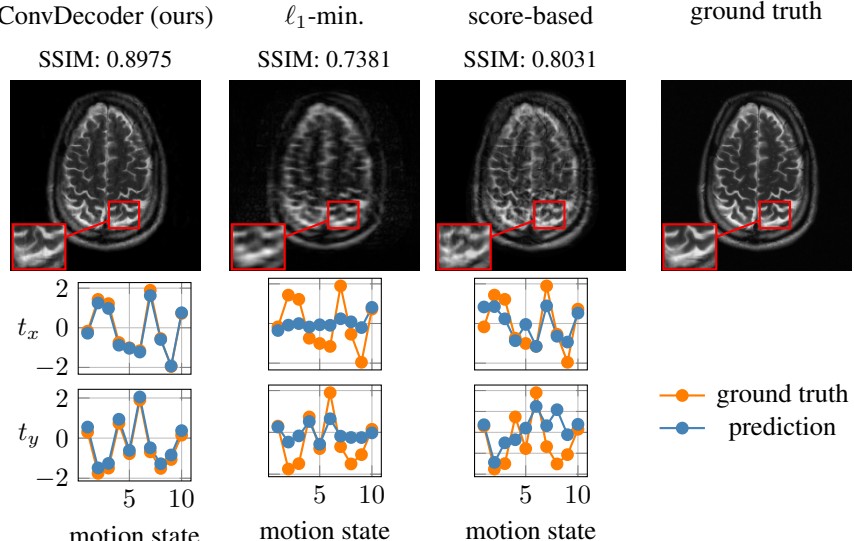

Figure 4: From the SSIM values and the reconstructions itself, we can see that our method outperforms $\ell_1$-minimization and score-based reconstruction methods. From the plots below which show the reconstructed motion parameters $t_x, t_y$ for each motion state, we can see that ConvDecoder performs best as it reconstructs the motion parameters better. Here, motion parameters are sampled from $\sim \text{Unif}(-2, 2)$ for each method and the acceleration factor is 4.

| pattern | #states | SSIM | | |
|---------|---------|------|---|---|
| | | ConvDecoder (ours) | $\ell_1$-min. | score-based |
| random | 10 | **0.8864** | 0.7406 | 0.7967 |
| random | 24 | **0.8831** | 0.7366 | 0.7643 |
| pseudo-realistic | 10 | **0.8824** | 0.7326 | 0.7612 |

Table 1: Our untrained network outperforms the $\ell_1$-minimization and score-based reconstruction algorithms for three motion pattern settings. SSIM scores are averaged over 336 AXT2 slices.

examples, we find the same ranking of algorithms as when ranking by SSIM in Table 1. Please see the supplement for further examples.

In terms of computational efficiency, our method takes approximately 6 minutes per slice (similar to $\ell_1$-minimization), whereas the score-based model takes approximately 30 minutes per slice. Runtimes were recorded on a single RTX A6000 GPU.

### 4.2.2 3D MOTION-COMPENSATED COMPRESSED SENSING MRI WITH UNTRAINED NETWORKS

A popular MRI protocol in practice that offers higher resolution is 3D volumetric MRI. As opposed to a 2D slice-by-slice measurement such as the fastMRI dataset (which we explored in the previous section), in volumetric MRI, there are two phase encoding dimensions.

Patient movements in 3D cause serious motion artifacts in volumetric MRI. In this section, we explain how our method can be applied to such 3D data and present an example reconstruction result. Our untrained network operates in a 2D space by default for the fastMRI dataset. To extend it to 3D, we simply replace every 2D operator by its 3D variant (e.g., replacing 2D convolutions by 3D convolutions). In this manner, the network generates a volume instead of a slice. An immediate consequence of this modification is a higher memory consumption and a larger inference time. Please see Table 2 for details.

To evaluate our method on a real-world clinically-recorded sample, we consider a 3D brain volume of size (#coil, H, W, D) = (31, 176, 176, 50). The volume is derived by downsampling a 3D Cartesian FLAIR scan recorded at a field strength of 3T with an original matrix size of (31, 704, 352, 281).

| data type | data size (#coils, H, W, D) | memory (GB) | runtime (mins) |
|-----------|------------------------------|-------------|----------------|
| 2D | (4, 640, 320, 1) | 2.1 | 6.3 |
| 3D | (31, 176, 176, 50) | 14.9 | 175.6 |

Table 2: Computational cost comparison between running our untrained network on a 2D or 3D sample. GPU memory and runtime numbers are reported for an RTX A6000 GPU.

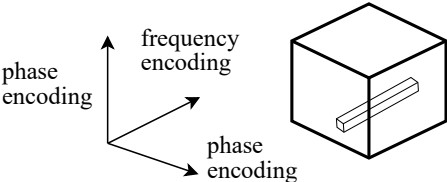

Figure 5: The 3D sampling trajectory type we consider in our 3D motion-compensated accelerated MRI reconstruction. Each readout along the frequency encoding direction is recorded via one excitation.

The 3D sampling trajectory using which the volume was recorded is shown in Figure 5. For motion artifacts, we considered 5 degrees of freedom: 3 rotations and 2 translations (we omitted $z$-axis translation (feet to head direction) as the patient's primary movement along this axis is expected to be nodding, which is already modelled by rotation).

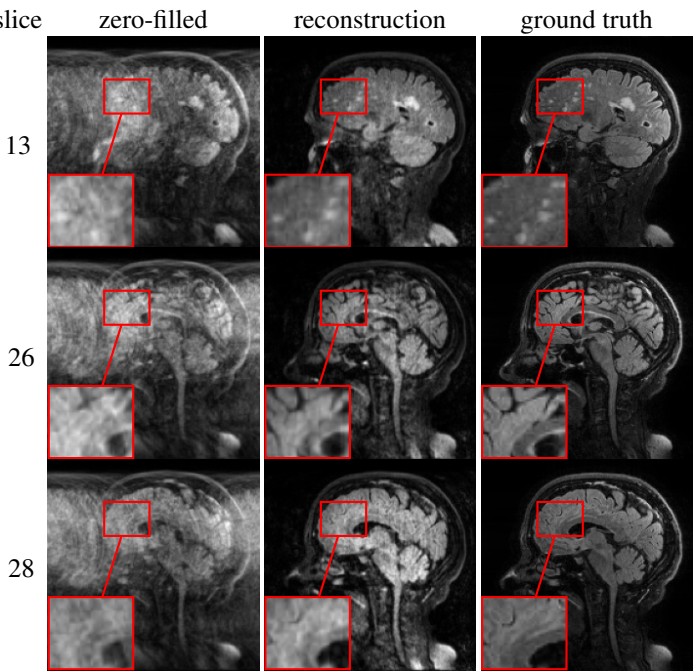

Figure 6: **3D untrained motion-compensated compressed sensing MRI.** Our qualitative analysis shows that for the depicted slices, an untrained network reconstructs a quality image.

To reconstruct the unknown ground truth volume, we fitted the network to the $2.4\times$ accelerated motion corrupted volume Figure 6 shows a few slices of the reconstructed 3D volume. We observe an amount of blurriness in all the reconstructed slices. Further, reconstructed slices 13 and 26 are of better quality in terms of the low amount of present motion artifacts, whereas slice 28 contains some residual artifacts.

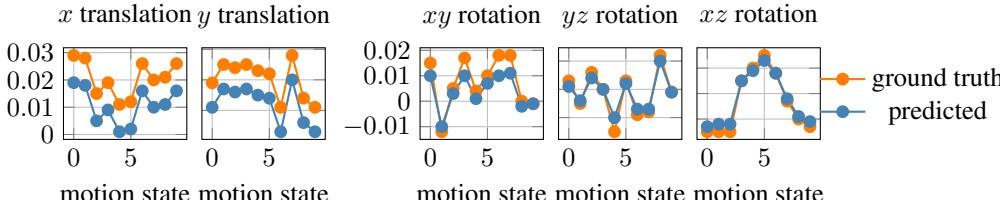

Figure 7: Our untrained network accurately recovers unknown motion transform parameters. Translation values are percentage of pixels and rotation values are in radian.

Finally in Figure 7, accurate recovery of motion parameters is shown. Note the offset between ground truth and predicted translation parameters which is due to the ambiguity of the reconstruction problem (i.e., a perfect reconstruction which is just a translated version of the ground truth image is still a valid solution to the problem).

## 5 DISCUSSION AND CONCLUSION

Deep learning achieves excellent performance in controlled scenarios for solving accelerated MRI reconstruction. However, in more realistic settings (such as accelerated MRI reconstruction from motion-degraded data), the performance and robustness of deep learning models is unclear.

In this work, we first demonstrated that state-of-the-art MRI reconstruction models become very expensive to use for motion-degraded MRI compressed sensing. This cost is reflected in the excessive amount of training data they require to achieve a similar performance compared to when they are employed for clean (artifact-free) MRI reconstruction.

We further proposed an approach based on untrained neural networks to solve the challenging task of motion-degraded compressed sensing MRI without any need for training data. Our method outperforms existing trained and untrained baselines w.r.t. to quantitative metrics as well as visual quality of the reconstruction.

Our work motivates further research in the direction of untrained network based motion-compensated compressed sensing MRI in multiple aspects. First, to study real-world (and not simulated) motion-degraded samples recorded with motion-recording sensors attached to the patient. Second, investigating the performance of trained and untrained networks under other important types of artifacts (e.g., respiratory artifacts). Finally, exploring the role of underasmpling trajectory in motion-degraded compressed sensing MRI and its effect on the performance of reconstruction models.

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

# A    RECONSTRUCTION EXAMPLES FOR VARIOUS MOTION PATTERN SETTINGS

Figure 8: Reconstruction examples and motion parameter plots given by our untrained network for three settings.

