# OpenReview forum: "Rigid Motion Compensated Compressed Sensing MRI with Untrained Neural Networks"
_ICLR.cc/2024/Conference — ICLR 2024 Conference Withdrawn Submission_

### Official Review · Reviewer_HZMo · 2023-10-30

**Soundness:** 3 good
**Presentation:** 2 fair
**Contribution:** 2 fair
**Rating:** 3
**Confidence:** 5

**Summary:**

The paper covers the task of MRI reconstruction in the presence of rigid motion. Several methods are considered, including supervised, untrained, and other approaches based on compressed sensing l1 minimization and score-based generative priors. The paper's experiments are in two sections. The first section looks at an idealized case for supervised approaches and shows that these methods give good performance in an idealized setting where motion is known, but can be greedy in terms of the required training data and simulated motions. The second set of experiments shows that untrained approaches (that also estimate the motion) outperform compressed sensing and score-based generative priors for the no-training data reconstruction task. The authors conclude that for the low-data regime, their extension of un-trained methods for rigid motion correction may be best for the task.

**Strengths:**

- The paper contains a detailed set of experiments for two classes of motion-compensated reconstruction: the first being an idealized experiment on supervised methods such as the E2E VarNet and the second being an extension of un-trained neural networks.
- The claim of supervised reconstruction requiring more training data seems qualified by Figures 2 and 3.
- The claim of un-trained methods being superior to l1-regularization and generative priors seems well-qualified by Table 1.
- The mathematics of rigid motion correction are clearly presented and lead to reasonable alterations of existing methods.

**Weaknesses:**

At the moment the paper does not seem complete. and so I am leaning towards rejection. A few points for my assessment are below:

1. The paper seems to propose un-trained methods as best for rigid motion-compensated reconstruction, but the SSIM numbers for the modified supervised methods are much higher for modified supervised methods (~0.88 for un-trained from Table 1, 0.92 for modified supervised in the high-data regime of Figure 3). The paper only considers an idealized setting for the supervised method and does not attempt to incorporate estimation of motion parameters for the idealized case. Why not simply propose the modified supervised method instead? In the abstract the paper's conclusion states, "Our approach outperforms untrained and trained optimization-based baselines such as ℓ1-norm minimization and score-based generative models," but this does not seem qualified with the paper own experiments.
2. The review of the motion-compensated reconstruction literature is fairly light, only including a couple of non-deep learning citations, despite this having been a major field in MRI reconstruction for the last few decades (with many products currently in scanners). Many of these methods are highly sophisticated in how they estimate the motion parameters, beyond tacking on a learnable module as done in Eq. (2) of the present paper.
3. Related to the above, the baselines in the paper may not be tailored for FSE acquisitions (particularly that based on generative priors) and as such could be underpowered for comparison.
4. The motion experiments considered in the paper are only simulated. Most MRI reconstruction publication venues would probably prefer to see the method qualified with prospective motion studies with subjects in a scanner.

**Questions:**

1. What is the exact reference for the l1 baseline used in the paper? Section 4.2.1 does not have a citation.
2. Why didn't the authors pursue an advanced method for estimating motion parameters to pair with the approach in Section 3?
3. Did you consider the use of navigator data? A small amount of navigator data may allow reliable estimation of the motion parameters, and this could be paired with the approach of Section 3.

---

### Official Review · Reviewer_DaWq · 2023-10-31

**Soundness:** 2 fair
**Presentation:** 3 good
**Contribution:** 2 fair
**Rating:** 3
**Confidence:** 4

**Summary:**

This paper extends deep image prior to motion compensated MRI reconstruction where an additional unknowns respect to patient motion are in the forward operator. The additional unknown is jointly optimized with the parameters of the neural network.

**Strengths:**

A simple but cute concept to extend DIP to motion compensated MRI reconstruction.

**Weaknesses:**

This idea is easy to follow. My main concerns are on the experimental setup and results.
- The results of E2EVarNet on motion-corrupted data needs more justification. In this paper’s setting, E2EVarNet perfectly knows the motion information, therefore E2EVarNet is equipped with the optimal forward model that can map the intermediate results to the k-space. That being said, the key idea of E2EVarNet, compared with end-to-end UNet, is its ability to compare against the intermediate reconstruction to the raw undersampled measurements. If the E2EVarNet has the optimal forward model, its performance shouldn’t be affected by motion. The forward operator with motion or without motion can both be expressed as a known operator.
- More classification on the score-based method. The score-based method is trained on a huge amount of motion-free MRI data (in general), then sample a new MRI from the learned distribution.  From this aspect, it seems unreasonable the score-based method would generate a MRI image with motion artifact (as in Figure 4), as the model has never been trained on such an image. Could author provide more discussion on this aspect. Also, how does the proposed method compared with this score-based method for MRI motion correction: Annealed Score-Based Diffusion Model for MR Motion Artifact Reduction, which doesn’t need to jointly estimate the motion information.
- The experimental setup can be improved. This paper only considered rigid and x4 acceleration factor, both are known to be less challenging. How does this method perform in more challenging case, such as non-rigid (also a common case in MRI) and x8/x10 acceleration factor? Moreover, l1-min baseline has a well-known weak non-learning-based prior. How does this baseline performance if equipped with a stronger learning-based prior, such as pre-trained denoisers?
- Lack of baseline: How does this method performance when comparing against the end-to-end method.

**Questions:**

See the weakness section.

**Details Of Ethics Concerns:**

In the introduction section, "we propose...." seems reveal the authorship of this paper, which breaks the blind submission policy.

---

### Official Review · Reviewer_SFXv · 2023-11-01

**Soundness:** 2 fair
**Presentation:** 3 good
**Contribution:** 2 fair
**Rating:** 5
**Confidence:** 4

**Summary:**

This paper describes an “untrained” neural network for MRI motion correction that does not require a dataset of motion-corrupted or motion-free images.

The paper first motivates the method via an experimental evaluation that shows that supervised networks trained to solve the combined undersampled reconstruction + motion correction task require significantly more data than the standard undersampled reconstruction task. As an alternative to learning from paired datasets, the proposed method uses a CNN to solve the task, where the network parameters are optimized for each test example via a data consistency loss that incorporates motion into the forward imaging model. This approach is evaluated on simulated motion-corrupted 2D MRI data as well as a single realistic 3D volume.

**Strengths:**

- To my knowledge, this is the first work that demonstrates “untrained” neural networks that do not require any paired datasets or ground-truth measurements in the context of the MRI motion correction problem, where the forward model is partially unknown and both the image and the motion parameters must be estimated.
- The results of the proposed method are promising.
- The experimental evaluation of test accuracy as a function of training dataset size for undersampled reconstruction and motion correction provides generally useful insights for the field of deep learning-based motion correction.

**Weaknesses:**

- **The experimental comparisons to a deep learning baseline are performed in an artificial simulation setting that ignores rotations, which is highly unrealistic and decreases the difficulty of the task.** Rotations are substantively different from translations in MR motion correction; while translations incur only a phase shift on the acquired data, rotations can result in missing “wedges” of the k-space data that yield qualitatively different artifacts. This makes it difficult to tell whether the observed trend of the proposed method outperforming the score-based generative model will hold in the more realistic setting where both rotations and translations are present.
- **More detailed results are needed to evaluate the method.** Please provide confidence intervals in Table 1 to give context for the differences between methods relative to the variability across the data. Also, the paper is missing a systematic quantification of the error in the estimated motion parameters — while these are shown for a single example in Figure 4, we need to see estimates of the motion parameter errors across the dataset to understand how well the method is learning the motion states (as opposed to just outputting a high-quality image without properly estimating the motion), and how this compares to other methods.
- **The realistic experiment on 3D data acquired with motion that does include rotations only compares to a zero-filled reconstruction, a very weak baseline that makes it difficult to understand how well the method is performing.** I am glad to see that rotations are modeled in the 3D motion model in section 4.2.2. However, the proposed method here is only compared to a zero-filled reconstruction, which is a very weak baseline, as this method does not even correct undersampling artifacts. At a minimum, the method should be compared to e.g. a GRAPPA reconstruction of the acquired image so the reader can understand the benefit provided by the proposed method compared to a standard reconstruction.

     I understand that supervised deep learning techniques can’t be compared to here, because large datasets of 3D motion corrupted MRI data are not easily available. But, another good comparison could be to a “grid of pixels” approach which uses the same loss function to optimize both the estimated motion parameters and the reconstruction pixel values themselves (with no network performing the reconstruction). I am guessing the proposed method would outperform such a baseline because of the prior induced by the convolutional architecture, but it’s important to show how much this improves performance or decreases runtime, as this is the core of the proposed method.
- **The paper is missing references to a large body of previous work in the ML for MRI motion correction literature.** It is worth noting explicitly in the text of Section 1.1 that this literature review focuses on retrospective brain MRI motion correction, as this section does not currently reference significant previous work on motion correction strategies that use motion measurements, prospective motion correction techniques, and non-rigid motion or motion in other commonly studied anatomies (e.g. cardiac imaging). Further, Section 1.1 is missing several references in the retrospective rigid brain MR motion correction literature; please consider reading and citing some subset of the following works.

    _Model-Based:_

     Cordero‐Grande, Lucilio, et al. "Three‐dimensional motion corrected sensitivity encoding reconstruction for multi‐shot multi‐slice MRI: application to neonatal brain imaging." Magnetic resonance in medicine 79.3 (2018): 1365-1376.

     Haskell, Melissa W., Stephen F. Cauley, and Lawrence L. Wald. "TArgeted Motion Estimation and Reduction (TAMER): data consistency based motion mitigation for MRI using a reduced model joint optimization." IEEE transactions on medical imaging 37.5 (2018): 1253-1265.

     _Data-Driven:_

     Duffy, Ben A., et al. "Retrospective motion artifact correction of structural MRI images using deep learning improves the quality of cortical surface reconstructions." NeuroImage 230 (2021): 117756.

     Johnson, Patricia M., and Maria Drangova. "Conditional generative adversarial network for 3D rigid‐body motion correction in MRI." Magnetic resonance in medicine 82.3 (2019): 901-910.

     Levac, Brett, et al. "FSE Compensated Motion Correction for MRI Using Data Driven Methods." International Conference on Medical Image Computing and Computer-Assisted Intervention. Cham: Springer Nature Switzerland, 2022.

     _Data-Driven and Model-Based:_

     Haskell, Melissa W., et al. "Network Accelerated Motion Estimation and Reduction (NAMER): Convolutional neural network guided retrospective motion correction using a separable motion model." Magnetic resonance in medicine 82.4 (2019): 1452-1461.

     Singh, Nalini M., et al. "Data Consistent Deep Rigid MRI Motion Correction." Medical Imaging with Deep Learning (2023).

- (minor) I think there is a small notation issue in the second line of Equation 1; the T operator previously was applied to image-domain data in the un-numbered equation on page 2, but here is operating on k-space data.

**Questions:**

- How does the proposed method compare to the baselines when both translations and rotations are modeled?
- What are the confidence intervals on the provided image reconstruction metrics?
- How well, quantitatively, does the proposed method estimate the motion parameters across the dataset and how does this compare to the baseline methods?
- How does the proposed method compare on the realistic 3D data to baselines that (1) correct the under sampling and (2) correct the motion correction in a similar setup to the current method, but without the convolutional architecture to guide the reconstruction?

---

### Official Review · Reviewer_tEj2 · 2023-11-05

**Soundness:** 3 good
**Presentation:** 3 good
**Contribution:** 2 fair
**Rating:** 3
**Confidence:** 4

**Summary:**

This paper proposes untrained neural networks for 2D and 3D accelerated MRI reconstruction under rigid motion during acquisition. First, a motion-compensated unrolled neural network (E2E-VarNet) is trained. Experiments are performed with varying training set size in the artifact-free regime, and the motion-corrupted regime, and it is shown that a large training set is needed in the motion-corrupted regime to attain a high reconstruction quality. Motivated by this observation, an untrained network for motion compensation is proposed and evaluated on 2D and 3D MRI, and is compared with recent baselines such as score-based models. The untrained network is shown to achieve superior performance on 2D MRI.

**Strengths:**

Strengths:

- The analysis on the scaling properties of end-to-end trained networks is an important contribution, illustrating the contrast in the scaling properties of adding more slices and adding more motion patterns to a fixed set of slices.
- The experiment for 2D MRI shows promising results, outperforming state-of-the-art baselines such as score-based models.

**Weaknesses:**

Weakness:

- The main weakness is the limited novelty of the paper. The application of untrained networks to the motion-corrupted regime is appreciated. However, the proposed method is a mere application of an existing technique with limited technical contribution.

- Another minor weakness is the limited scope for experiments. A comparison with a competitive baseline is provided only in the 2D case with a single dataset, and a single acceleration rate which makes it difficult to quantify the generalizability of the results. This weakness could be easily alleviated by adding more testing scenarios (e.g. more undersampling rates, datasets), or by adding a competitive baseline in the 3D case (e.g. l1-min, score-based).

**Questions:**

Questions:

-  "The parameters of a network that learns motion parameters ϕ are trained". Can you provide more details about this procedure? How is $T_{\phi}$ trained?
- Why is only SSIM included as a quality metric? Please also include PSNR or another image quality metric, as it is difficult to quantify results with a single metric.
- How does the performance of the modified E2E-VarNet compare with on the test set considered in the 2D experiments shown in Table 1?
- How do baselines perform in the 3D case? Including these would improve the experiments section of the paper.